



# The impact of unimolecular reactions on the possibility of acyl peroxy radical initiated isoprene oxidation

Ida Karppinen[1], Dominika Pasik[1,2], Emelda Ahongshangbam[1,2], and Nanna Myllys[1,2]

[1]Department of Chemistry, University of Helsinki, Helsinki, 00014, Finland
[2]Institute for Atmospheric and Earth System Research, University of Helsinki, Helsinki, 00014, Finland

**Correspondence:** Nanna Myllys (nanna.myllys@helsinki.fi)

**Abstract.** The unimolecular H-shift and endoperoxide ring formation reactions were studied for several different acyl peroxy radicals (APRs) using quantum-mechanical methods. Also, for structures with slow unimolecular reactions, accretion reactions with isoprene were investigated. The reaction rate coefficients were calculated at the DLPNO-CCSD(T)/aug-cc-pVTZ//$\omega$B97X-D/6-31+G* level using multi-conformer transition state theory. Unimolecular reactions of acyl peroxy radicals were shown to have rate coefficients up to 0.1 s$^{-1}$ and bimolecular accretion reactions with isoprene up to $10^{-15}$ cm$^3$s$^{-1}$. Both smaller and larger acyl peroxy radicals with rigid structures were observed to be more likely to initiate accretion reactions with isoprene because of their inability for fast unimolecular reactions. The pseudo-first-order reaction rates were calculated for accretion reactions of isoprene with OH and six APRs at four different temperatures. The significance of ARP-initiated isoprene oxidation was shown to increase with increasing temperature. APR-initiated oxidation could lead to dimeric products with atmospheric impact through new-particle formation.

## 1 Introduction

Peroxy radicals (RO$_2$) can undergo a variety of different reactions in the atmosphere. Bimolecular reactions with hydroxyl radicals, other RO$_2$ radicals and nitrogen oxides are common for RO$_2$ radicals (Goldman et al., 2021). Unimolecular reactions, such as H-shift reactions and endoperoxide ring formation reactions, are also possible reaction pathways for RO$_2$ radicals. An interesting class of RO$_2$ radicals are acyl peroxy radicals (APRs), which have shown to be more reactive than other RO$_2$ radicals (Knap and Jørgensen, 2017; Seal et al., 2023; Møller et al., 2019; Vereecken and Nozière, 2020; Nozière and Fache, 2021). APRs are generally formed from aldehydes when the aldehydic hydrogen atom is removed by photolysis or in a reaction with the OH radical (Demore et al., 1997; Atkinson et al., 1997). In these reactions, an acyl radical forms and rapidly reacts with O$_2$ forming an APR (Atkinson and Arey, 2003). They can also be formed by photolysis of ketones (El-Agamey and McGarvey, 2002) or indirectly from aldehydes through a process called autoxidation (Crounse et al., 2013; Knap and Jørgensen, 2017). However, the direct APR formation route from aldehydes is the dominant one, since the removal of an aldehydic H atom is much faster than the removal of a non-aldehydic H atom (Barua et al., 2023).

A recent study by Pasik et al. (2024a) showed that APRs, unlike other RO$_2$ radicals, are enough reactive towards double bonds that these reactions can occur under atmospheric conditions. Accretion reaction of an APR to an unsaturated hydrocarbon





leads to a dimeric product with an alkyl radical center (Nozière et al., 2023). $O_2$ can add to this radical center forming new $RO_2$ radicals which may undergo autoxidation. The APR-initiated oxidation described leads to compounds with high molecular mass and multiple oxygen atoms. These compounds have low vapor pressure and therefore can potentially participate in atmospheric new-particle formation (NPF). NPF accounts for a major part of tropospheric aerosol production which can act as cloud condensation nuclei (CCN) and also negatively affect human health (Lee et al., 2019). However, it has also been found

that APRs may go through fast H-shift reactions (Knap and Jørgensen, 2017; Møller et al., 2019; Vereecken and Nozière, 2020; Seal et al., 2023) making their reactions with unsaturated hydrocarbons less likely. Also, APRs with a double bond in the structure can go through endoperoxide ring formation reactions (Nozière and Vereecken, 2024). These reactions have been studied for $RO_2$ radicals and shown to compete with H-shift reactions (Vereecken and Peeters, 2004; Xu et al., 2019; Møller et al., 2020). Vereecken et al. (2021) also studied some APR ring closure reactions, and they showed that APR exhibits faster

ring closure reactions compared to $RO_2$ radicals.

The unimolecular H-shift and ring closure reactions of APR leads to the formation of an alkyl radical to which $O_2$ rapidly adds forming a peroxy radical. It means that if APR undergoes fast unimolecular reactions, it cannot react with unsaturated hydrocarbons. Therefore, these unimolecular reactions have to be slow enough to allow accretion reactions to double bond which has been illustrated in Figure 1. The aim of this study is to find APR structures with slow ($k_{uni} \leq 10^{-4}\mathrm{s}^{-1}$) unimolecular

reactions but fast ($k_{bi} > 10^{-17}\mathrm{cm}^3\mathrm{s}^{-1}$) bimolecular accretion reactions to double bond. Hydroxyl radical (OH) accounts for 66-95 % of isoprene oxidaton (Gu et al., 2022), and therefore, the APR-initiated oxidation rate is compared to that of OH. Moreover, the temperature dependence of the bimolecular reactions is investigated, and the pseudo-first-order reaction rate coefficients are presented at different temperatures.

## 2  Methods

### 2.1  Computational details

Transition state (TS) structures were obtained using the relaxed potential energy surface (PES) scan using density functional theory (DFT) level $\omega$B97X-D/6-31+G* (Chai and Head-Gordon, 2008b, a; Hehre et al., 1972; Clark et al., 1983). For unimolecular reactions, TS structures were located starting from the reactant, whereas for bimolecular reactions, they were located starting from the product. Prior to the PES scan, the reactant and product structures were optimized at the $\omega$B97X-D/6-31+G*

level of theory. Reactant and TS conformers were found using the CREST sampling tool at the GFN2-xTB level (Pracht et al., 2020; Bannwarth et al., 2019). TS conformer configurational search included constraining the bonds forming the TS structure. For unimolecular H-shift reactions, this involved constraining the C−H and H−O bond distances, while for endoperoxide ring formation reactions, the O−C bond being formed was constrained. For bimolecular reactions the C−O bond being broken was kept constrained. Reactant and TS conformers were optimized at the same DFT level as previously, using a 2.5 kcal/mol

cutoff after the CREST configurational search, as xTB level electronic energies correlate well with DFT energies (Pasik et al., 2024b; Kubečka et al., 2019). Duplicates were removed based on electronic energy and dipole moment, and TS structures were confirmed by one imaginary frequency. For the lowest energy reactant and TS structures, single point energies were calculated





**Figure 1.** Possible reaction pathways for acyl peroxy radicals.

using the DLPNO-CCSD(T)/aug-cc-pVTZ level of theory (Riplinger and Neese, 2013; Riplinger et al., 2013; Dunning Jr, 1989; Kendall et al., 1992). All DFT calculations were carried out using Gaussian 16 software, and ORCA version 5.0.3 was

used for the single-point energy calculations (Frisch et al., 2016; Neese, 2012).

To compare the significance of APR-initiated isoprene oxidation to that of OH, we performed calculations for the reactions between isoprene and OH as well. The TS structures for the bimolecular reactions between OH and isoprene could not be found with the $\omega$B97X-D functional. This is because the saddle point is very shallow or might not even exist (see Supplementary Information Figures S2 and S3 for PES graphs). Thus, alternatively, the M06-2X functional was used (Zhao and Truhlar, 2008),

and all reactant and TS conformers were optimized at the M06-2X/6-31+G* level for the reactions between OH and isoprene. On top of that, single-point energies were calculated for the lowest energy conformers using the DLPNO-CCSD(T)/aug-cc-pVTZ level of theory. Three of the studied bimolecular reactions between isoprene and APRs (ace-APR reaction R4, pro-APR reaction R1, and ben-APR reaction R1) were additionally investigated at the DLPNO//M06-2X level to examine how the rate coefficients differ from those calculated at the DLPNO//$\omega$B97X-D level. There is only a minor difference between zero-

point corrected electronic energies at the studied DLPNO//DFT levels. However, partition functions between those density functionals differ significantly, which can affect an order of magnitude difference in rate coefficients. The results from these benchmark calculations are provided in the Supplementary Information (Table S2).



## 2.2 Rate coefficients

The multi-conformer transition state theory (MC-TST) was utilized to calculate the reaction rate coefficients (Vereecken and
Peeters, 2003). The rate coefficients for the unimolecular reactions were calculated using Equation 1 (Møller et al., 2016).

$$k = \kappa_{\text{t}} \frac{k_{\text{B}}T}{h} \frac{\sum_{i}^{\text{allTSconf.}} \exp(-\frac{\Delta E_i}{k_{\text{B}}T}) Q_{\text{TS},i}}{\sum_{j}^{\text{allRconf.}} \exp(-\frac{\Delta E_j}{k_{\text{B}}T}) Q_{\text{R},j}} \exp(-\frac{E_{\text{TS}} - E_{\text{R}}}{k_{\text{B}}T}), \tag{1}$$

where $k_{\text{B}}$ is the Boltzmann's constant, $T$ is the temperature and $h$ is the Planck's constant. $\Delta E_i$ is the zero-point corrected
energy of TS conformer $i$ relative to the lowest energy TS conformer and $Q_{\text{TS},i}$ is the partition function of TS conformer $i$
both calculated at the $\omega$B97X-D/6-31+G* level. $\Delta E_j$ and $Q_{\text{R},j}$ are the analogous values for the reactant conformers. $E_{\text{TS}}$ and
$E_{\text{R}}$ are the zero-point corrected energies of the lowest energy TS and reactant conformer, respectively, including the DLPNO-
CCSD(T)/aug-cc-pVTZ correction.

$\kappa_{\text{t}}$ is the quantum-mechanical tunneling coefficient which was calculated using the one-dimensional Eckart tunneling method
(Eckart, 1930; Johnston and Heicklen, 1962). Tunneling was needed for H-shift reactions due to the low mass of the hydrogen
atom (McMahon, 2003). To calculate the tunneling coefficient, forward and reverse intrinsic reaction coordinate (IRC) calcu-
lations were carried out to connect the lowest energy TS to the corresponding reactant and product (Møller et al., 2016). The
resulting reactant and product structures were optimized at the $\omega$B97X-D/6-31+G* level followed by DLPNO-CCSD(T)/aug-
cc-pVTZ energy corrections. Additionally, the imaginary frequency of the lowest energy TS was utilized to calculate the
tunneling coefficient.

The rate coefficients for the bimolecular reactions were calculated using Equation 2 (Viegas, 2018, 2021).

$$k = \kappa_{\text{t}} \frac{k_{\text{B}}T}{h P_{\text{ref}} Q_{\text{ip}}} \frac{\sum_{i}^{\text{allTSconf.}} \exp(-\frac{\Delta E_i}{k_{\text{B}}T}) Q_{\text{TS},i}}{\sum_{j}^{\text{allRconf.}} \exp(-\frac{\Delta E_j}{k_{\text{B}}T}) Q_{\text{R},j}} \exp(-\frac{E_{\text{TS}} - E_{\text{R}}}{k_{\text{B}}T}), \tag{2}$$

where $P_{\text{ref}}$ is the reference pressure ($= 2.45 \times 10^{19}$ molecules cm$^{-3}$) and $Q_{\text{ip}}$ is the partition function of isoprene. Only the
partition function of the lowest energy conformer of isoprene was used due to the rigid structure.

## 3 Results

The rates of unimolecular H-shift and endoperoxide ring formation reactions for several different APR structures are calcu-
lated. In addition, for structures exhibiting slow unimolecular reaction rates, bimolecular accretion reactions with isoprene are
investigated. The APR structures and their corresponding names used in this study are presented in Figure 2. These structures
include not only small APRs but also larger cyclic APR structures that are potentially too rigid for fast unimolecular reactions.







**Figure 2.** The chemical structures of acyl peroxy radicals investigated in this study. The fastest unimolecular reactions are marked with dots, green for H-shifts and blue for endoperoxide ring formations. For clarification, the prefixes for structures with two reactions with similar rates are also marked.

### 3.1 Unimolecular reactions

We calculated the reaction rate coefficients for unimolecular H-shift and endoperoxide ring formation reactions. For clarity, Table 1 presents only the results for the fastest calculated rate coefficients. The fastest reactions are also illustrated in Figure 2. Other calculated rate coefficients can be found in the Supplementary Information (Table S1).

The slowest unimolecular reactions were observed for the smallest systems such as ace-APR and cyc3-APR. The optimiza-
tion of the TS structure for the 1,4 H-shift reaction of ace-APR turned out to be difficult. A decomposition of the product results
in the formation of ethenone and hydroperoxy radical during the optimization, making it difficult to find the correct TS. This decomposition channel of the product is presented in Figure 3. The reaction has also been studied in more detail by Sandhiya and Senthilkumar (2020). However, after several attempts, the correct TS structure was found and the correct rate coefficient was calculated for this H-shift reaction.

**Figure 3.** 1,4 H-shift reaction of ace-APR and the decomposition of the product.

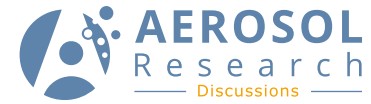

**Table 1.** Calculated energy barrier heights ($\Delta E^{\mathrm{TS}}$ in kcal/mol), Eckart tunneling coefficients ($\kappa_{\mathrm{t}}$) for H-shifts and unimolecular MC-TST reaction rate coefficients ($k_{\mathrm{uni}}$ in s$^{-1}$) at 298 K of fastest unimolecular reactions for the studied APRs. The APRs that have slow ($k_{\mathrm{uni}} \leq 10^{-4}\,\mathrm{s}^{-1}$) unimolecular reactions are bolded.

| Radical | Reaction | $\Delta E^{\mathrm{TS}}$ | $\kappa_{\mathrm{t}}$ | $k_{\mathrm{uni}}$ |
|---|---|---|---|---|
| **Ace-APR** | 1,4 H-shift | 29.79 | 460 | $1.07 \times 10^{-7}$ |
| **Pro-APR** | 1,5 H-shift | 24.61 | 100 | $4.49 \times 10^{-5}$ |
| Iso-APR | 1,4 H-shift | 22.95 | 43 | $3.49 \times 10^{-3}$ |
|  | 1,5 H-shift | 22.99 | 87 | $2.92 \times 10^{-3}$ |
| Piv-APR | 1,5 H-shift | 22.47 | 37 | $1.32 \times 10^{-3}$ |
| **Acr-APR** | 4-endoperoxide | 22.75 | - | $3.15 \times 10^{-5}$ |
|  | 5-endoperoxide | 23.66 | - | $3.95 \times 10^{-6}$ |
| Met-APR | 1,5 H-shift | 24.30 | 1249 | $1.61 \times 10^{-3}$ |
|  | 4-endoperoxide | 20.36 | - | $1.51 \times 10^{-3}$ |
| **Ben-APR** | 4-endoperoxide | 24.80 | - | $1.94 \times 10^{-6}$ |
|  | 5-endoperoxide | 23.55 | - | $5.67 \times 10^{-6}$ |
| Pyro-APR | 1,5 H-shift | 25.50 | 25302 | $4.77 \times 10^{-3}$ |
|  | 4-endoperoxide | 19.66 | - | $6.18 \times 10^{-3}$ |
| Cyc6-APR | 1,5 H-shift | 19.95 | 14 | $5.06 \times 10^{-2}$ |
| Cyc5-APR | 1,5 H-shift | 20.44 | 50 | $7.65 \times 10^{-2}$ |
| Cyc4-APR | 1,5 H-shift | 22.62 | 43 | $1.46 \times 10^{-3}$ |
| **Cyc3-APR** | 1,5 H-shift | 29.40 | 18 | $4.13 \times 10^{-9}$ |
| Pin1-APR | 1,6 H-shift | 23.54 | 92 | $1.32 \times 10^{-3}$ |
| Pin2-APR | 1,5 H-shift | 19.84 | 44 | $1.60 \times 10^{-1}$ |
| **Naf1-APR** | 5-endoperoxide | 20.86 | - | $3.25 \times 10^{-4}$ |
| Naf2-APR | 6-endoperoxide | 17.42 | - | $5.81 \times 10^{-2}$ |

By increasing system size an increase in the unimolecular rate coefficients was observed which was to be expected. Larger
system size allows H-shifts from further positions relative to the peroxy radical group resulting in a larger TS ring size with less
strain. However, slower reactions were also observed for rigid systems, for example, ben-APR and naf1-APR. This is due to the
rigid structure and aromaticity of the benzene ring which does not allow fast H-shift nor endoperoxide ring formation reactions.
However, naf2-APR is capable of quite fast unimolecular reactions despite the rigid structure. By adjusting the position of the
APR group, a 6-endoperoxide ring formation reaction becomes possible, leading to the formation of a 6-membered ring product
and facilitating a faster unimolecular reaction. A similar reaction mechanism is not efficient for naf1-APR, as the formation of
the 6-membered ring is hindered.

Endoperoxide ring formations were found to be competitive with H-shift reactions or, in most cases, even faster. Unexpectedly, 4-endoperoxide reactions for acr-, met-, and pyro-APR were faster than the 5-endoperoxide reactions. Contrastingly,



Vereecken et al. (2021) studied a similar peroxy radical system without the acyl group and concluded that the 5-endoperoxide

reaction was faster than the 4-endoperoxide reaction. This result for APRs is possibly due to the $\pi$-bond being delocalized between the acyl group and the double bond, which may favor the 4-endoperoxide reaction. The TS structure of acr-APR 4- and 5-endoperoxide reactions are presented in Figure 4. However, exact conclusions can not be made only based on the calculated energies and this is only speculation. Further investigation on the structures would be needed to better understand this trend.

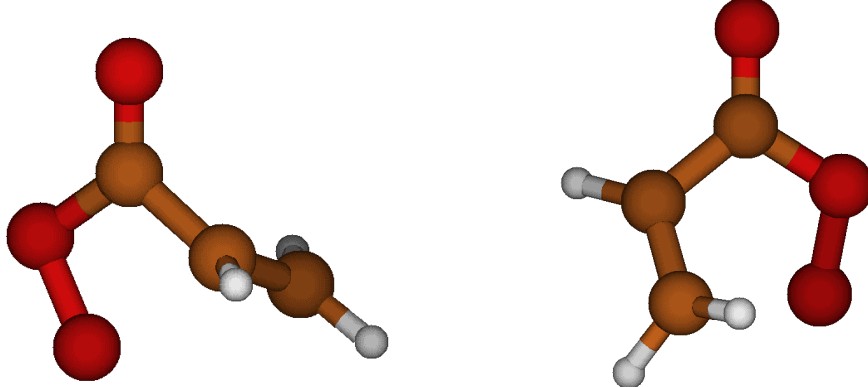

**Figure 4.** Transition state structures of acr-APR 4-endoperoxide (left) and 5-endoperoxide (right) ring formation reaction. Color coding: brown is carbon, red is oxygen, and white is hydrogen.

An allylic hydrogen migration appears to be faster than the H-abstraction at an alkyl site, an observation consistent with

other $RO_2$ radicals as well (Vereecken and Nozière, 2020). This trend can be seen in met-APR allylic 1,5 H-shift in comparison to the similar non-allylic 1,5 H-shift in iso-APR. The 1,5 H-shift in met-APR is two orders of magnitude faster than the 1,5 H-shift in iso-APR which can be explained by the resonance stabilization in the product of the 1,5 H-shift in met-APR. In this reaction, a relatively high barrier of 21.25 kcal/mol is overcome by a large tunneling coefficient of 1249. This same trend can be observed for a similar allylic 1,5 H-shift in pyro-APR. The reaction has a high barrier of 25.50 kcal/mol but also a very

large tunneling coefficient of 25302 which results in a faster reaction than expected from the barrier. However, this represents a significant tunneling factor, suggesting that the Eckart method may no longer be reliable in this case.

### 3.2    Bimolecular reactions

Isoprene and peroxy radicals can react through four different pathways, where the peroxy radical adds to one of the four $sp^2$ carbons in isoprene. These four pathways are presented in Figure 5. Pasik et al. (2024b) studied these different pathways and

showed that reaction pathways R1 and R4, where the peroxy radical adds to one of the two terminal $sp^2$ carbons, are the fastest. The reaction pathway R1 leads to a tertiary (allylic) radical and R4 to a secondary (allylic) radical which makes them faster than reaction pathways R2 and R3 which lead to primary radicals. Therefore, only reaction pathways R1 and R4 were the focus of this study.

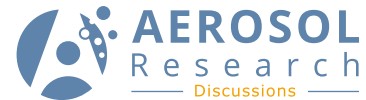



**Figure 5.** Reaction pathways for reaction between isoprene and acyl peroxy radical.

Of interest were the APR structures which had slow ($k_{uni} \leq 10^{-4}\text{s}^{-1}$) unimolecular reactions. In correspondence, six APR structures (ace-, pro-, acr-, ben-, cyc3-, and naf1-APR) were chosen for further calculations to study their bimolecular reactions (see Table 1). The same reaction pathways were also studied for the reaction between isoprene and OH to assess the significance of APR-initiated oxidation of isoprene. Calculations on reactions between isoprene and APRs were carried out at the DLPNO-CCSD(T)/aug-cc-pVTZ//$\omega$B97X-D/6-31+G* level and calculations for OH reactions at the DLPNO-CCSD(T)/aug-cc-pVTZ//M06-2X/6-31+G* level. Table 2 presents the calculated rate coefficients for both reaction pathways R1 and R4. The table also includes the forward reaction barrier heights and the total reaction rates which were assumed to be a sum of reaction rates R1 and R4.

As expected, reaction pathway R1 is the fastest for all studied structures. This is explained by the reaction product having a tertiary resonance stabilised carbon-centered radical in comparison to reaction pathway R4 which leads to a secondary (allylic) radical. It can also be seen that the forward barrier decreases as reactant APR size increases, aligning with the observations made by Pasik et al. (2024b). The larger reactant size results in a larger reaction product which stabilizes the structure. An exception to this is naf1-APR reaction R1 which has a slightly larger barrier than the corresponding reaction of ben-APR despite the larger product structure of naf1-APR reaction. However, the barrier for naf1-APR reaction R1 is still close to zero resulting in a relatively fast reaction. Also, when comparing reactions R1 and R4 for pro- and acr-APR, it can be noted that the forward barriers are approximately 1 kcal/mol lower for acr-APR than for pro-APR. The TS and product structures for acr-APR are more stabilized due to the double bond in the acr-APR structure which results in lower forward barriers and faster reactions compared to pro-APR which does not have a double bond in the structure.





**Table 2.** Calculated energy barrier heights ($\Delta E^{\text{TS}}$ in kcal/mol) and bimolecular MC-TST reaction rate coefficients ($k_{\text{bi}}$ in $\text{cm}^3\text{s}^{-1}$) at 298 K for reactions R1 and R4 of isoprene with OH and APRs. Rates for APR and isoprene reactions were calculated at the DLPNO-CCSD(T)/aug-cc-pVTZ//$\omega$B97X-D/6-31+G* level and for OH and isoprene reactions at the DLPNO-CCSD(T)/aug-cc-pVTZ//M06-2X/6-31+G* level. Total $k_{\text{bi}}$ is the sum of reaction rates R1 and R4.

| Radical | Reaction | $\Delta E^{\text{TS}}$ | $k_{\text{bi}}$ | Total $k_{\text{bi}}$ |
|---|---|---|---|---|
| Ace-APR | R1 | 1.8 | $1.2 \times 10^{-16}$ | $1.3 \times 10^{-16}$ |
| | R4 | 2.8 | $1.1 \times 10^{-17}$ | |
| Pro-APR | R1 | 1.3 | $1.3 \times 10^{-16}$ | $1.3 \times 10^{-16}$ |
| | R4 | 2.3 | $9.5 \times 10^{-18}$ | |
| Acr-APR | R1 | 0.4 | $7.1 \times 10^{-16}$ | $7.8 \times 10^{-16}$ |
| | R4 | 1.4 | $7.4 \times 10^{-17}$ | |
| Ben-APR | R1 | 0.1 | $1.6 \times 10^{-15}$ | $1.7 \times 10^{-15}$ |
| | R4 | 1.0 | $1.2 \times 10^{-16}$ | |
| Cyc3-APR | R1 | 0.6 | $2.2 \times 10^{-16}$ | $2.6 \times 10^{-16}$ |
| | R4 | 1.6 | $3.3 \times 10^{-17}$ | |
| Naf1-APR | R1 | 0.5 | $7.7 \times 10^{-16}$ | $9.9 \times 10^{-16}$ |
| | R4 | 0.7 | $2.2 \times 10^{-16}$ | |
| OH | R1 | -2.5 | $3.2 \times 10^{-11}$ | $5.0 \times 10^{-11}$ |
| | R4 | -2.4 | $1.8 \times 10^{-11}$ | |

An experimental value of $1.0 \times 10^{-10}$ $\text{cm}^3\text{s}^{-1}$ (Atkinson et al., 1997) has been obtained for the reaction between OH and isoprene which differs from our theoretical value by a factor of two. The value calculated in this study does not take into account the other two reaction pathways R2 and R3 which do also contribute to the rate coefficient. The values for the three reactions of isoprene with ace-, pro-, and ben-APR calculated at the DLPNO-CCSD(T)/aug-cc-pVTZ//M06-2X/6-31+G* level are provided in the Supplementary Information (Table S2). The rate coefficients calculated with the M06-2X functional are smaller, up to approximately an order of magnitude smaller, compared to the ones calculated at the DLPNO-CCSD(T)/aug-cc-pVTZ//$\omega$B97X-D/6-31+G* level. Therefore, the different levels of theory used in this study are not completely compatible with each other and make the comparison of OH- and APR-initiated isoprene oxidation more difficult. However, reasonable conclusions can be made from the results and the impact of APR-initiated oxidation of isoprene can be assessed.

The change of bimolecular reaction rates as a function of temperature was also investigated at the same levels of theory as the initial calculations. Rate coefficients were calculated at four different temperatures (248 K, 273 K, 298 K, and 323 K) and the total bimolecular rates were again assumed to be a sum of reaction rates R1 and R4. The pseudo-first-order reaction rates were calculated from the total bimolecular rates using a concentration of $10^6$ $\text{cm}^{-3}$ for OH (Wennberg et al., 2018) and a concentration of $10^8$ $\text{cm}^{-3}$ for all APRs. The APR concentration used is a mean value of ace-APR concentration in hydrocarbon-rich remote atmospheres under low $\text{NO}_x$ conditions from a study by Villenave et al. (1998). Data on other APR




concentrations could not be obtained and the ace-APR concentration was used as a default concentration for all other APRs. This assumption leads to some uncertainty in the results but they can be assumed to be an upper limit for the pseudo-first-order rates. The results are provided in Table 3. An increase in the pseudo-first-order rates for the APR reactions can be observed as temperature increases. On the contrary, a decrease in the pseudo-first-order rate of reaction between isoprene and OH can be distinguished as temperature increases. The negative temperature dependence of OH-initiated oxidation of isoprene has also been observed in experimental studies (Dillon et al., 2017; Kleindienst et al., 1982). This leads to a larger impact of APR-initiated oxidation of isoprene compared to that of OH at higher temperatures. At 248 K APRs could be responsible for up to 0.1% of isoprene oxidation compared to OH-initiated isoprene oxidation and at 323 K the significance of APR-initiated oxidation could be up to 1%.

**Table 3.** Calculated pseudo-first-order reaction rates ($k_{\text{pseudo}}[\text{radical}]$ in $\text{cm}^{-3}\text{s}^{-1}$) for studied reactions of isoprene with OH and APRs in four different temperatures ($T$ in K). Concentrations of $10^6$ molecules $\text{cm}^{-3}$ and $10^8$ molecules $\text{cm}^{-3}$ were used for OH and all APRs, respectively.

| T | $k_{\text{pseudo}}[\text{Ace-APR}]$ | $k_{\text{pseudo}}[\text{Pro-APR}]$ | $k_{\text{pseudo}}[\text{Acr-APR}]$ | $k_{\text{pseudo}}[\text{Ben-APR}]$ | $k_{\text{pseudo}}[\text{CycPro-APR}]$ | $k_{\text{pseudo}}[\text{Naf1-APR}]$ | $k_{\text{pseudo}}[\text{OH}]$ |
|---|---|---|---|---|---|---|---|
| 248 | $5.9 \times 10^{-9}$ | $7.3 \times 10^{-9}$ | $5.2 \times 10^{-8}$ | $1.3 \times 10^{-7}$ | $1.6 \times 10^{-8}$ | $6.4 \times 10^{-8}$ | $1.2 \times 10^{-4}$ |
| 273 | $9.1 \times 10^{-9}$ | $1.0 \times 10^{-8}$ | $6.4 \times 10^{-8}$ | $1.5 \times 10^{-7}$ | $2.1 \times 10^{-8}$ | $8.1 \times 10^{-8}$ | $7.5 \times 10^{-5}$ |
| 298 | $1.3 \times 10^{-8}$ | $1.3 \times 10^{-8}$ | $7.8 \times 10^{-8}$ | $1.7 \times 10^{-7}$ | $2.6 \times 10^{-8}$ | $9.9 \times 10^{-8}$ | $5.0 \times 10^{-5}$ |
| 323 | $1.9 \times 10^{-8}$ | $1.8 \times 10^{-8}$ | $9.5 \times 10^{-8}$ | $2.0 \times 10^{-7}$ | $3.2 \times 10^{-8}$ | $1.2 \times 10^{-7}$ | $3.6 \times 10^{-5}$ |

## 4 Conclusions

We investigated unimolecular H-shift and endoperoxide ring formation reactions for a variety of acyl peroxy radicals using quantum-mechanical methods. We selected 16 APR structures with different functionalities. As expected, many of the studied APRs had quite fast unimolecular reactions, up to $0.1$ s$^{-1}$. However, we found several APRs with low unimolecular reaction rate coefficients. For instance, small ace- and pro-APRs have reaction rate constants in the order of $10^{-7}$ and $10^{-5}$ s$^{-1}$, respectively. An even slower reaction was observed for a rigid APR with a 3-membered ring, cyc3-APR, for which the reaction rate constant was found to be as low as $10^{-9}$ s$^{-1}$. For aromatic structures, the endoperoxide ring formation was faster than the H-shift reaction, but due to rigid structure and aromaticity, the reaction rate constants were still low, in order of $10^{-5}$ s$^{-1}$ for ben-APR and $10^{-4}$ s$^{-1}$ for naf1-APR. Another naphthalene structure, naf2-APR, was able to form a 6-membered endoperoxide ring, and therefore, the unimolecular endoperoxide ring formation reaction was fast. Generally, where endoperoxide ring formation reactions were possible, they were shown to be competitive with H-shift reactions, and in most cases even faster.

Bimolecular accretion reactions between isoprene and APRs were also investigated in this study. Six APRs with slow unimolecular reactions were chosen for further calculations. Two pathways producing an allylic radical, R1 and R4, were studied for reactions between isoprene and the APRs. Pathway R1 was proven to be faster due to the formation of a tertiary allylic radical. Quite consistently, the reaction barrier heights decreased as system size increased for both pathways. The temper-

ature dependence of these reactions was also studied and compared to OH-initiated isoprene oxidation. We calculated the pseudo-first-order reaction rates for isoprene oxidation initiated by OH and six APRs at four different temperatures. For all studied APRs, the reaction rate increased as temperature increased, whereas in the case of OH, the reaction rate decreased as temperature increased. This indicates a larger impact of APR-initiated isoprene oxidation compared to that of OH at higher

temperatures. At 323 K, APRs could be responsible for up to 1% of isoprene oxidation in the atmosphere compared to OH. While the percentage of APR-initiated oxidation of isoprene compared to OH is not significant, APR-initiated oxidation can directly lead to high molar mass dimeric products. These dimeric compounds with multiple oxygen atoms and high molar mass have low vapor pressure, and therefore, they are candidates to participate in new-particle formation and growth. Thus, despite the minor contribution to initiate oxidation, APRs might have an important role in producing low-volatility organic compounds.

For a better understanding of the impact of APR-initiated oxidation, measurements contributing to APR concentrations in the atmosphere are needed.

*Data availability.* The optimized structures and calculation output files of all relevant compounds that support the findings of this manuscript will be available in the Zenodo repository.

*Author contributions.* IK performed the calculations and wrote the manuscript. DP and EA assisted with the calculations. DP and NM

contributed to the analysis. The study was designed and supervised by NM. All authors proofread the manuscript.

*Competing interests.* The authors declare that they have no conflict of interests.

*Acknowledgements.* We acknowledge the Research Council of Finland for funding (grant nos. 347775) and the CSC-IT Center for Science in Espoo, Finland, for computational resources. DP thanks the Doctoral Programme in Atmospheric Sciences (ATM-DP) at the University of Helsinki for providing funding.



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
