# Peer review of "The impact of unimolecular reactions on acyl peroxy radical initiated isoprene oxidation"

_Aerosol Research, 2024_

## Author Response (AR1)

Dr. Nanna Myllys
Academy Research Fellow
Docent in Atmospheric Science
Department of Chemistry
University of Helsinki
00014 UH Finland
+358503812141
nanna.myllys@helsinki.fi

March 10, 2025

Dear Jonas,

We hereby submit the revised paper entitled: "The impact of unimolecular reactions on acyl peroxy radical initiated isoprene oxidation" for publication in *Aerosol Reserach.* We thank all reviewers for their suggestions and insights. Manuscript has been revised considering the comments brought up by the reviewers, which has further clarified the message of this study and improved the quality of the text. We hope that the following responses are satisfying and that the paper can be accepted for publication in AR.

Sincerely,

**Nanna Myllys**

HELSINGIN YLIOPISTO
HELSINGFORS UNIVERSITET
UNIVERSITY OF HELSINKI

MATEMAATTIS-LUONNONTIETEELLINEN TIEDEKUNTA
MATEMATISK-NATURVETENSKAPLIGA FAKULTETEN
FACULTY OF SCIENCE

We thank all reviewers for their insightful comments. Please find below a point-by-point replies to the comments. Modifications made to the manuscript are visible in the tex file.

Referee 1: Comments to the Author

This manuscripts describes a quantum chemical and theoretical kinetic study of acylperoxy radicals (APR), examining both unimolecular reactions (H-migrations and ring closures), and addition reactions on the double bonds in isoprene. This topic is timely as there is significant interest at the moment for reactions forming low-volatility oxygenates in the atmosphere as contributors to aerosols. The accretion reactions of APR with olefines would lead to such low-volatility oxygenates with further options for autoxidation. The methodology is state of the art and expected to yield reliable results; the rate predictions are directly usable in modeling studies.

I support publication of this paper, but suggest the authors consider the comments below.

REPLY: We thank the reviewer for their comprehensive comments and have made changes according to them to the manuscript. Please find our line-by-line responses to your comments below.

Main comments

The literature cited in this paper suggest that the reactions of APR are a recent topic. However, I feel the literature cited in this work does not do justice to the already existing body of data on APR reactions. The unimolecular and bimolecular reactions of APR have been studied for decades, and they were not included in atmospheric models because they have a negligible impact on the chemistry of RO2 and VOCs. The re-kindled interest in these reactions stems mostly from the realization that organic aerosol formation and growth is governed by reactions with yields in the single-percentage range, and even less (see e.g. HOM yields from monoterpenes). See below for some suggestions of additional relevant literature; a comparison to earlier experimental data might be in order.

REPLY: Thank you for your comment and the suggestions. We have considered the literary cited in the manuscript, made some changes and added new references based on the suggestions in later comments. We have also added comparisons to earlier data. Changes made to the manuscript are shown in later comments where these references and suggestions from the reviewer are discussed in more detail. New references are DOI 10.5194/acp-19-7691-2019, DOI 10.1093/oso/9780199767076.001.0001, DOI 10.1039/d1sc02263f, DOI 10.1021/jp972054+, DOI 10.1023/A:1005940332441 and DOI 10.1021/jp993612i.

Modern measurements of total RO2 concentrations in the atmosphere yield values around 1E8 cm-3, of which CH3O2 is the main contributor. This study assigns a concentration of 1E8 cm-3 to all APR, even the more exotic ones like Ben-APR and Naf1-APR with the highest rate coefficients with isoprene. At the same time, the study uses a low rate coefficient for isoprene + OH, where the latest IUPAC recommendation, k(T) = 2.1E-11 exp(465/T), with [OH]=1E6 cm-3 yields a pseudo-first order rate k(323K) = 9E-5 s-1, roughly 3 times higher than the value used by the authors. Combined, these choices provide a very generous "upper limit" for the impact of APR accretion reactions of 1% at 323 K; my own estimates would be at least an order of magnitude lower. Both still confirm earlier conclusions that such accretion reactions have negligible impact on both RO2 and VOC in the atmosphere. However, that also misses the point of why these reactions are studied: what is interesting is whether these reactions can contribute meaningfully to low-volatility HOMs formation. To assess that, the authors need to look at the expected yields from the studied reactions, against the measured yield of HOMs from the most common VOCs (isoprene but also a-pinene, limonene,...) which typically is only a few % of the turnover even for the most favorable ones. I suggest the authors include such a comparison.

REPLY: Thank you for this very valuable insight. We agree that the proposed upper limit of 1% at 323 K for the impact of APR accretion reactions is generous but we also want to emphasize that it indeed is an upper limit. Experimental data presented by Nozière and Fache 2021 (DOI 10.1039/d1sc02263f) showed measured rate constants for ace-APR in order of $10^{-14}$ cm$^3$ s$^{-1}$ which is two orders of magnitude larger than

our calculations. Thus, while our calculated absolute rate constants are too small, the relative APR vs. OH rate constant ratio is likely to be fine.

We also want to point out that the goal of this study is not to underline the impact of APR-initiated isoprene oxidation but to determine which APRs could work as oxidants of unsaturated hydrocarbons. Then, reactions of APRs with other unsaturated hydrocarbons, such as monoterpenes, might have bigger impacts on atmospheric chemistry. The goal of our study and atmospheric implications are now also more clearly stated in the manuscript.

 Minor comments

p. 1, Title: The subphrase "the possibility of" could be removed.

REPLY: The title has been changed to "The impact of unimolecular reactions on acyl peroxy radical initiated isoprene oxidation".

p. 1 line 12: "biomolecular reactions with hydroxyl radicals".

While RO2 + OH does contribute somewhat to RO2 loss (see work by Assaf and Fittschen), it is mostly the HO2 (hydroperoxyl) forming ROOH that is relevant in the atmosphere. When it comes to atmospheric chemistry of RO2, perhaps the overview paper by Jenkin et al. might be a good alternative reference (DOI 10.5194/acp-19-7691-2019)

REPLY: Thank you for your insight. This part in the manuscript has been changed according to the suggestions given.

The following has been added to the manuscript:

"\ce{RO2} reactions with \ce{NO_x} and \ce{NO3} are important in moderately to highly polluted conditions, however in clean environments, reactions with other species, mainly \ce{HO2}, become more significant \citep{jenkin2019estimation}."

p. 1, line 21: "However, the direct APR formation route from aldehydes is the dominant one, since the removal of an aldehydic H atom is much faster than the removal of a non-aldehydic H atom (Barua et al., 2023)."

This has been known for a long time. Barua et al. is not the optimal reference for that statement.

REPLY: Thank you for pointing this out. A new reference, DOI 10.1093/oso/9780199767076.001.0001, better suited for this statement has been added.

p. 1, line 23: "A recent study by Pasik et al. (2024a) showed that APRs..."

Rate data has been available for much longer. This needs a reference to Nozière and Fache 2021, as well as the rate data in Stark et al 1997 (irrespective of the expected products), and there is more literature available.

REPLY: Thank you for these suggestions. The references mentioned were added to the manuscript.

The following has been changed to the manuscript:

"Studies have shown that APRs, unlike other \ce{RO2} radicals, are enough reactive towards double bonds that these reactions can occur under atmospheric conditions \citep{noziere2021reactions, pasik2024gas}. Experimental studies of \ce{RO2} gas-phase reactions with alkenes in high temperatures have also shown that APRs are more reactive towards double bonds than other \ce{RO2} \citep{stark1997epoxidation}."

p. 2, line 33: "...through endoperoxide ring formation reactions (Nozière and Vereecken, 2024)."

Another paper of interest could be DOI 10.1039/d1cp02758a, 2021, and perhaps other work cited therein. It would also be beneficial to compare somewhere the APR ring closure reactions described here against the

olefinic RO2 ring closure described in the suggested paper to quantify the effect of the acyl group on these reactions.

REPLY: A reference to the suggested paper was added. This same reference is also mentioned later in the same paragraph for comparing $RO_2$ ring closure reactions to that of APR. In addition, the values calculated in the suggested paper were compared to the ones obtained in our study in the results part.

The following was added to the manuscript:

"\cite{vereecken2021structure} studied ring closure reactions and calculated rate coefficients for multiple different \ce{RO2} radicals, including \ce{C2H3CH2OO}$\cdot$, a non-acyl peroxy radical system similar to acr-APR. For \ce{C2H3CH2OO}$\cdot$, values of $4.6\times10^{-11}$ $\text{s}^{-1}$ and $2.3\times10^{-10}$ $\text{s}^{-1}$ were obtained for 4- and 5-endoperoxide reactions, respectively, in the study by \cite{vereecken2021structure}. The values obtained in this study for acr-APR are significantly higher compared to \ce{C2H3CH2OO}$\cdot$, suggesting that ring formation reactions are faster for APRs compared to other \ce{RO2}."

p. 3, line 63: "This is because the saddle point is very shallow or might not even exist (see Supplementary Information Figures S2 and S3 for PES graphs)."

The reaction of OH with double bonds is typically described as a 2-TS system where a pre-reactive complex is formed first (the H-atom facing the double bond). E.g. Greenwald et al, 2005, DOI 10.1021/jp058041a and especially 10.1021/jp071412y are relevant. Comparison to this earlier work, and subsequent work by other authors, is necessary. At least it needs to be stated explicitly whether wB98X-D gives a pre-reactive complex or not.

REPLY: It depends on the method used, wB97X-D vs. M06-2X, if we are able to find the TS or a pre-reactive complex. We have added the energy diagram for reaction R1 between OH and isoprene in the Supplementary Information (Figure S4). From the energy diagram, it can be seen that the complex and TS are close in energy which is why it is hard to find the correct TS and also the pre-reactive complex.

p. 3, line 64: "the M06-2X functional"

This is a bit an unfortunate choice. For long-range interactions such as low-barrier additions, dispersion should have been included to get the lowest modes correct (as well as e,g, better integration grids), as these modes are the most influential on the partition function. I.e. M06-2X-D3 might have been a better choice, and methodologically better comparable to wB98X-D which also includes dispersion. I recognize it is too late to change this, but perhaps these shortcomings in the methodology choice could be mentioned somewhere.

REPLY: The M06-2X is extensively parametrized, and the empirical parameters make the functional account for medium-range correlation contribution. While it is possible to apply additional dispersion corrections with M06-2X (the zero-damping D3), it is not very common. M06-2X-D3 might lead to overbinding and thus the results are not necessarily improved. This is why we did not feel it was needed to add the dispersion correction in the original calculations. However, we did additional calculations including the D3-dispersion correction on top of the M06-2X functional and calculated rates for the reactions originally calculated with the M06-2X functional. Results from these calculations differed from the benchmark calculations presented in the Supplementary Information only very slightly meaning that the addition of the dispersion correction did not make the results better comparable to wB97X-D.

p. 4, line 92: "Only the partition function of the lowest energy conformer of isoprene was used due to the rigid structure."

This reasoning is not always valid. Example: Decker et al. 2017, DOI 10.1039/C6CP08602K

REPLY: The second isoprene conformer is about 1.5 kcal/mol higher in energy than the lowest energy conformer. This leads to a small exponential factor of 0.08 in the MC-TST equation and thus in a minor contribution to the overall rate which is why we have only taken into account the lowest energy conformer of isoprene. This was also clarified in the manuscript.

The following was added to the manuscript:

"The second isoprene conformer is 1.5 kcal/mol higher in energy than the lowest energy conformer which would lead to a small exponential factor of 0.08 in Eq. \ref{rate_bi} and a small contribution to the overall rate. Therefore, only the partition function of the lowest energy conformer of isoprene was considered in Eq. \ref{rate_bi}."

p. 7, line 120: "This result for APRs is possibly due to the $\pi$-bond being delocalized between the acyl group and the double bond, which may favor the 4-endoperoxide reaction."

In the TS, the double bonds seem to be not co-planar and thus little to no delocalization seems possible; if anything, the loss of the conjugation of the C=C-C=O bonds in the TS would increase the barrier. No speculation is necessary though: if there is delocalization it should be borne out by the population analysis in the calculations. However, the impact of the different geometric properties of the endocyclic C=O bond, and the presence of an oxygenated group adjacent to the addition site seem more likely contributors than delocalization.

REPLY: Thank you for your insight. The speculation has been removed all together.

p. 9, line 158: "The value calculated in this study does not take into account the other two reaction pathways R2 and R3 which do also contribute to the rate coefficient."

Mention that these contributions are only in the % range (with appropriate reference)

REPLY: The following was added to the manuscript:

"However, these contributions are minor, somewhere in the \% range \citep{jenkin1998peroxy,stevens2000theoretical}."

Referee 2: Comments to the Author

This manuscript investigated unimolecular H-shift and endoperoxide ring formation reactions for a variety of acyl peroxy radicals using quantum-mechanical methods. Bimolecular accretion reactions between isoprene and APRs were also investigated in this study. The temperature dependence of these reactions was also studied and compared to OH-initiated isoprene oxidation. The pseudo-first-order reaction rates were calculated for accretion reactions of isoprene with OH and six APRs at four different temperatures. The significance of APR-initiated isoprene oxidation was shown to increase with increasing temperature. APR-initiated oxidation could lead to dimeric products with atmospheric impact through new-particle formation. I would support this manuscript for publication at *Aerosol Research* if the authors could carefully address the following comments.

REPLY: We thank the reviewer for valuable feedback, and we have clarified the manuscript accordingly. Please find our line-by-line responses to your comments below.

1. Line 8: "ARP" should be "APR".

REPLY: This was corrected

2. In the introduction section, the authors introduced the research background and the current research progress, which is comprehensive and easy to follow. However, the difficulties and limitations of existing research are not shown well. Additionally, the significance of the study and its potential environmental impacts are not clearly articulated. These elements should be more explicitly stated to highlight the broader implications of the research.

REPLY: Thank you for pointing out the deficiencies in the introduction. We have modified the introduction to make the motivation behind this study clearer to the reader. The following was added to the manuscript: "Previous studies on unimolecular reactions of APRs have mainly focused on which of these reactions are fast in atmospheric conditions." -- "Our goal is to determine what type of APRs could work as oxidants of unsaturated hydrocarbons and initiate the formation of low volatility compounds. These APRs with slow unimolecular reactions and fast bimolecular accretion reactions to double bond are potential oxidants of unsaturated hydrocarbons."

3. The M06-2X functional was used, and all reactant and TS conformers were optimized at the M06-2X/6-31+G* level for the reactions between OH and isoprene. Why was the M06-2X functional chosen for the reactions between OH and isoprene? It is recommended that the authors provide further explanation and clarify why this functional is more suitable.

REPLY: The saddle point for this reaction is very shallow and therefore it was difficult to find a suitable functional to study this reaction with. With M06-2X, we were able to find the TS structure and this is why this method was implemented. Other functionals and basis sets were also used but the method chosen was the only one that worked for this reaction in our case. It is mentioned in the manuscript that simply because the saddle point for the reaction is shallow/does not exist, we chose another functional because with M06-2X we were able to obtain the TS structure. We have also presented the PES scans in the Supplementary Information to showcase the differences for OH and isoprene reactions between the two functionals used in our study. We have now also added energy diagrams to the Supplementary Information which show that the TS energy is close to the energy of a pre-reactive complex which also explains why the TS was difficult to find.

4. It is suggested that the authors analyze the uncertainty of the theoretical calculation results, including parameter and model errors, to evaluate the reliability of the results.

REPLY: Compared to measurements conducted by Nozière and Fache 2021 (DOI 10.1039/d1sc02263f), our APR bimolecular rate constants are approximately two orders of magnitude lower. We are also underestimating the rate for OH + isoprene reaction compared to experimental values, and therefore, the relative rates should be reliable. We have added the comparison and analysis of the uncertainty of the results to the manuscript:

"Both (experimental) rates are considerably higher than the total rate calculated in this study. This would suggest that the rates calculated in this study underestimate the total bimolecular rates for reactions between

APRs and isoprene. We are also underestimating the rate constant for OH reaction with isoprene. An experimental value of $1.0\times10^{-10}$ $\text{cm}^3$ $\text{s}^{-1}$ \citep{atkinson1997evaluated} has been obtained for the reaction between OH and isoprene which differs from our theoretical value by a factor of two." -- "While we are underestimating the rates for these reactions, the comparison between OH- and APR-initiated isoprene oxidation should be reliable."

5. Line 104: A decomposition of the product results in the formation of ethenone and hydroperoxy radical during the optimization, making it difficult to find the correct TS. How was the correct TS structure attempted to be found during the calculation process?

REPLY: The correct TS structure was attempted to be found by scanning from both the reactant and the product. We also did constrained optimizations but ultimately the correct TS structure was obtained by optimizing an already existing TS structure provided in DOI: 10.1039/d3cp01833d.

6. This result for APRs is possibly due to the π-bond being delocalized between the acyl group and the double bond, which may favor the 4-endoperoxide reaction. Please consider citing other recent work or adding clarification.

REPLY: Due to comments from another reviewer, this speculation was removed all together. According to referee 1, no delocalization would seem possible in this case, making our speculation invalid.

7. Line 126: The 1,5 H-shift in met-APR is two orders of magnitude faster than the 1,5 H-shift in iso-APR. How is the difference of two orders of magnitude obtained?

REPLY: This part has been removed from the manuscript due to changes in results and this statement not being correct anymore. We could not observe a difference in rate coefficients between met- and iso-APR.

8. Line 160: Is it possible to further quantify the relationship between barriers and reaction rates for different APR structures? For example, is there any quantitative barrier difference or other data to support this trend?

REPLY: It seems that large and unsaturated APR structures lead to the lowest barriers, and thus, highest reaction rates. It could be that especially unsaturated APRs can effectively stabilize the breaking of the pi bond. Similar trends have also been observed for other RO$_2$ radicals by Nozière and Fache 2021 (DOI 10.1039/d1sc02263f). Based on this, modifications have been made and the following is stated in the manuscript:
"It can also be seen that the forward barrier decreases as reactant APR size increases. Unsaturated APR, acr-APR, also exhibits faster addition reactions to isoprene. These results align with observations made in other studies for APRs and other \ce{RO2} radicals \citep{pasik2024cost,noziere2021reactions}. Unsaturated and large APRs could effectively stabilize the breaking of the $\pi$ bond as the electron density can be delocalized over several atoms resulting in decreased barriers."

9. The discussion on the changes in reaction rates at different temperatures was very interesting, especially that the APR reaction rate increases with increasing temperature, while the OH-initiated

isoprene reaction rate decreases with increasing temperature. However, the mechanism of the effect of temperature on the reaction rate was not sufficiently elaborated.

REPLY: Addition of OH to the double bond has a negative barrier whereas APR has a positive barrier. Therefore, OH addition has a negative temperature dependence and APR addition a positive temperature dependence. The following was added to the manuscript:

"This difference in the temperature dependence can be attributed to differences in the barriers. The addition of OH to isoprene has a negative barrier, whereas APR additions have positive barriers."

Referee 3: Comments to the Author

This manuscript uses electronic structure theory and multi-conformation transition state theory to explore the possibility that acyl peroxy radicals (APRs) can undergo bimolecular accretion reactions with isoprene to form adducts that may contribute to atmospheric new particle formation. This study thus has high potential relevance to atmospheric chemistry. The crux of the study is to compare the pseudo-first-order rate constants for accretion to the rate constants for unimolecular APR reactions and to the pseudo-first-order rate constants for isoprene-OH reactions. The electronic structure methods used are reasonable, with the need for, and consequences of, using both ωB97X-D and M06-2X density functionals clearly explained. The results are presented very clearly. I have only two suggestions about the interepretation of the results:

REPLY: We thank the reviewer for pointing out parts in the manuscript needing more explanation and consideration. The manuscript has been modified to better answer these points. Please find our line-by-line responses to your comments below.

1. On p. 7, explain why a very large tunneling factor (~$10^4$) may make the Eckart prediction not reliable.

REPLY: We have considered your comment and the following is now stated in the manuscript:
"For pyro-APR 1,5 H-shift a very large tunneling coefficient of 25302 was observed. A large imaginary frequency of $2238i~\text{cm}^{-1}$ results in a large tunneling coefficient. This coefficient differs from other calculated tunneling coefficients significantly, which could suggest that the Eckart method may no longer be reliable in this case. The Eckart method is very simple compared to multidimensional models that also take into account other variables along the reaction path \citep{zhang2011impact, meana2011high}. However, for the purposes of this study the Eckart method is reasonable due to the saved computational resources."

2. On p. 8, the trend of decreasing activation barrier with larger APR needs more consideration. First, what is relevant is the stability of the transition state (TS) structure, not the stability of the reaction product, as stated in line 150. Second, what controls the activation barrier is the energetic cost of starting to break a pi bond in isoprene. A bigger APR may indeed be more stable, but that

additional stability is present in both the reactant and the TS structure--the difference in reactant and TS energy, which is the activation barrier--should be roughly the same.

REPLY: Thank you for pointing this out. We have considered the trend again and based on this, made modifications to the manuscript and the following is now stated in the manuscript:

"It can also be seen that the forward barrier decreases as reactant APR size increases. Unsaturated APR, acr-APR, also exhibits faster reactions with isoprene. These results align with observations made in other studies for APRs and other \ce{RO2} radicals \citep{pasik2024cost,noziere2021reactions}. Unsaturated and large APRs could effectively stabilize the breaking of the $\pi$ bond as the electron density can be delocalized over several atoms resulting in decreased barriers."